# Robotic Parenchymal-Sparing Pancreatectomy: A Systematic Review

**DOI:** 10.3390/cancers15174369

**Published:** 2023-09-01

**Authors:** Richard Zheng, Elie Ghabi, Jin He

**Affiliations:** Department of Surgery, Johns Hopkins Hospital, Johns Hopkins University, Baltimore, MD 21287, USA

**Keywords:** robotic, minimally invasive, pancreatectomy, pancreas preserving, parenchymal preserving, parenchymal sparing, enucleation, central pancreatectomy

## Abstract

**Simple Summary:**

Non-anatomic pancreatic resections such as enucleation, duodenum-preserving partial pancreatic head resection, central pancreatectomy, and uncinate resection allow for the preservation of more pancreatic parenchyma than standard resections, i.e., Whipple and distal pancreatectomy. These lead to a significantly lesser degree of endocrine and exocrine insufficiency. Robotic approaches are increasingly being adopted for these technically challenging parenchymal-sparing procedures. The aim of our study was to evaluate the use and added value of the robotic approach compared to open approaches and standard anatomic resections. We carried out a systematic review of the available literature surrounding robotic parenchymal-sparing pancreatectomy and found that while postoperative pancreatic fistula remains common, severe complications are exceedingly rare, and rates of endocrine and exocrine insufficiency are negligible after these procedures.

**Abstract:**

Background: Parenchymal-sparing approaches to pancreatectomy are technically challenging procedures but allow for preserving a normal pancreas and decreasing the rate of postoperative pancreatic insufficiency. The robotic platform is increasingly being used for these procedures. We sought to evaluate robotic parenchymal-sparing pancreatectomy and assess its complication profile and efficacy. Methods: This systematic review consisted of all studies on robotic parenchymal-sparing pancreatectomy (central pancreatectomy, duodenum-preserving partial pancreatic head resection, enucleation, and uncinate resection) published between January 2001 and December 2022 in PubMed and Embase. Results: A total of 23 studies were included in this review (*n* = 788). Robotic parenchymal-sparing pancreatectomy is being performed worldwide for benign or indolent pancreatic lesions. When compared to the open approach, robotic parenchymal-sparing pancreatectomies led to a longer average operative time, shorter length of stay, and higher estimated intraoperative blood loss. Postoperative pancreatic fistula is common, but severe complications requiring intervention are exceedingly rare. Long-term complications such as endocrine and exocrine insufficiency are nearly nonexistent. Conclusions: Robotic parenchymal-sparing pancreatectomy appears to have a higher risk of postoperative pancreatic fistula but is rarely associated with severe or long-term complications. Careful patient selection is required to maximize benefits and minimize morbidity.

## 1. Introduction

The traditional approach to the resection of any pancreatic mass involves pancreaticoduodenectomy in the case of right-sided lesions and a distal pancreatectomy and splenectomy for left-sided lesions. Even in high-volume centers, these operations have a significant rate of complications, such as pancreatic fistula, intra-abdominal infection, hemorrhage, and delayed gastric emptying. More recently, parenchymal-sparing pancreatectomy (PSP) has become an accepted approach in the management of indolent and premalignant pancreatic pathologies such as pancreatic cystic lesions and well-differentiated pancreatic neuroendocrine tumors (PanNET), as minimizing morbidity and disability to the patient is crucial. Techniques vary based on the location of the lesion. These include duodenum-preserving partial pancreatic head resection (DPPPHR) for lesions in the head of the pancreas, uncinate resection for uncinate process lesions, central pancreatectomy for lesions in the body, and enucleation for right-sided lesions that do not block the pancreatic duct. These procedures allow for the preservation of normal pancreas parenchyma, which may reduce the exocrine and endocrine insufficiency that is commonly seen with a more extensive resection. However, these are technically challenging operations and are not performed at all centers, let alone performed via a minimally invasive approach.

Although many surgeons have increasingly carried out laparoscopic pancreatectomies, the learning curve is incredibly steep. Robotic pancreatic surgery, however, is an emerging technique that has become increasingly performed since the first robotic-assisted pancreaticoduodenectomy by Giulianotti in 2001 [1,2]. The Da Vinci robotic platform allows for improved visualization, wrist articulation, and precise manipulation. These advancements should allow for the performance of the technically demanding parenchymal-preserving operations in a safer, more efficient fashion. However, whether we are delivering on the promise of decreasing morbidity by performing robotic parenchymal-preserving operations is unknown.

We aim to summarize the current literature surrounding robotic parenchymal-sparing pancreatectomy (rPSP) and compare outcomes of rPSP with open parenchymal-sparing operations.

## 2. Methods

### 2.1. Literature Search Strategy

Two authors (R.Z. and E.G.) independently searched the literature for studies on rPSP published between January 2001 and December 2022 in PubMed (Medline) and Embase. The search strategy and study protocol were registered on INPLASTY (INPLASY202340032). IRB approval was not required to conduct the systematic review. The comprehensive list of used search terms and the search strategy are available in the Appendix A.

### 2.2. Study Design and Quality Assessment

This systematic review was performed per the Preferred Reporting Items for Systematic Review and Meta-Analysis (PRISMA) guidelines [3]. Non-randomized comparative studies were assessed using the Methodological Index for the Non-Randomized Studies (MINORS) tool [4]. Case reports and case series were assessed using the Joanna Briggs Institute (JBI) critical appraisal checklists for case reports and case series, respectively [5].

### 2.3. Inclusion and Exclusion Criteria

All studies were independently reviewed by two authors (R.Z. and E.G.). After duplicates were removed, titles and abstracts of all identified studies were screened. Studies were excluded if both authors agreed. Conflicts were resolved by a third author (J.H.) and by consensus. Following the title and abstract screening phase, full-text screening was performed for all included studies. Only studies published in the English language were selected. Inclusion criteria were as follows: (1) studies that report the use of a rPSP technique, (2) studies that compare outcomes of a rPSP technique to a specific comparator group, and (3) studies with sufficient data regarding the outcomes of interest. The exclusion criteria were (1) editorials and letters to the editor, (2) review articles, and (3) studies where data pertinent to the rPSP group/subgroup could not be reliably extracted. Techniques for rPSP included DPPPHR, uncinate resection, central pancreatectomy, and enucleation.

### 2.4. Data Extraction

Data extraction was performed by E.G. and crosschecked by R.Z. Numbers were extracted from the original manuscripts for binary and categorical variables. The mean and standard deviation were extracted from the original manuscript when available for continuous variables. Medians and (interquartile) ranges were converted to means and standard deviations using the formulas described by Wan et al. [6].

### 2.5. Outcomes of Interest

Primary outcomes of interest were postoperative exocrine and/or endocrine insufficiency, overall and severe morbidity (severe morbidity was defined as greater than Clavien–Dindo (CD) grade 3), and risk of postoperative pancreatic fistula (POPF). Secondary outcomes of interest were short-term operative outcomes; these include operative time, length of hospital stay, estimated blood loss, number of transfusions, rate of conversion to open surgery, and need for reoperation.

## 3. Results

### 3.1. Search Results and Study Evaluation

A total of 422 articles were identified (PubMed *n* = 135, Embase *n* = 287). After duplicates were removed (*n* = 46), 376 articles underwent title and abstract screening, and 75 articles were considered for full-text assessment. Only 23 studies were included after full-text assessment for this review.

All studies were published between 2010 and 2022. The studies include six case reports, [7,8,9,10,11,12] six case series, [13,14,15,16,17,18] and eleven comparative studies [19,20,21,22,23,24,25,26,27,28,29], of which one was a randomized clinical trial [22]. Case reports and series are summarized in Table 1, while retrospective studies are summarized in Table 2. Studies predominantly originated from Asia (*n* = 14). A total of 799 patients were identified, encompassing 788 unique patients and 11 patients reported from the same authors in an earlier institutional case series.

All case reports and case series were evaluated using the JBI critical appraisal checklist and were found to be of acceptable quality. The median MINORS score for comparative studies was 16. One retrospective study was given a score of 14 [27]. The randomized controlled trial reported by Chen et al. was given a score of 20 [22]. Among the comparative studies, three studies had unique control groups. One study compared robotic enucleation (rEN) to robotic pancreatoduodenectomy/distal pancreatectomy [20]. Another compared a robotic end-to-end pancreatic anastomosis to pancreatojejunostomy (PJ) reconstruction [18]. The last study compared robotic central pancreatectomy (rCP) in elderly patients versus young patients [29]. The remaining comparative studies compare rEN to open EN (oEN) [21,23,25,27] and rCP to open CP (oCP) [19,22,24,26].

### 3.2. Demographic and Clinical Characteristics

Among all included studies, the mean age for patients undergoing rPSP was 48.9 years. Males made up 37.2% of the total study sample. Body mass index (BMI) was reported in 14 out of 23 studies with a mean value of 22.6 kg/m^2^ [13,14,15,18,20,21,22,23,23,25,26,27,27,30]. ASA status was not consistently reported; when reported, patients with an ASA status <3 comprised 91.2% of the study sample [10,11,13,15,16,18,20,21,22,25,25,29,30].

For patients who underwent rEN, a mean age of 50.1 years and a mean BMI of 25.2 kg/m^2^ were observed. Patients with ASA status <3 comprised 88.2% of the subgroup. Similarly, for patients who underwent rCP, a mean age of 48.6 years and a mean BMI of 23 kg/m^2^ were observed and patients with an ASA status <3 comprised 92.5% of the subgroup.

When including only case reports and case series, there were a total of 48 patients with a mean age of 47.6 years and a mean BMI of 19.6 kg/m^2^. Half (*n* = 24/48, 50%) of the tumors were located in the pancreatic neck/body; 16.7% (*n* = 8) were in the head, 10.4% (*n* = 5) in the tail, 2% (*n* = 1) in the uncinate, and the remainder were in an unknown location. The size of these tumors ranged from 1 to 3.9 cm. The most common pathology among these case reports and case series was PanNET (45.8%, *n* = 22), followed by SCN (14.6%, *n* = 7), MCN (12.5%, *n* = 6), and SPN (18.8%, *n* = 9). Additionally, there were single cases of IPMN, mass-forming pancreatitis, and incidentally-discovered metastatic renal cell carcinoma.

When including only retrospective cohort studies and randomized trials, there was a total of 751 subjects (with 473 of these being rPSP cases). After further stratifying these into comparative studies that compared rEN to oEN and rCP to oCP, there were 596 subjects included. Among this subcohort, age [19,21,22,23,24,25,26,27], BMI [21,22,23,25,26,27], and ASA status [21,22,25] were not significantly different between the two subgroups. 

### 3.3. Surgical Indications and Operative Characteristics

Preoperative evaluation was limited and inconsistently reported in the included studies; however, most studies indicated that a rPSP technique was selected to manage a benign or benign-appearing lesion. Overall, masses in the proximal pancreas comprised 15.5% of lesions and included 10.9% of tumors in the pancreatic head and 4.6% in the uncinate process. Neck/body lesions comprised of 75.4% of all lesions, while tail lesions comprised 7.9%. Tumors included 32.4% PanNETs, 15.4% intraductal papillary mucinous neoplasms (IPMN), 14.4% solid pseudopapillary neoplasms (SPN), and 9.8% mucinous cystic neoplasms (MCN). There were two cases of incidentally-discovered malignant lesions (0.3%) which comprised metastatic hepatocellular carcinoma, and one case of metastatic renal cell carcinoma. The case of metastatic hepatocellular carcinoma was initially diagnosed preoperatively as an SPN or PanNET. The case of metastatic renal cell carcinoma was initially thought to be a benign tumor or a tumor with low-malignant potential on preoperative imaging. The overall median tumor size for rPSP was 2.3 cm. Among the comparative studies that included an open PSP group, rPSP median tumor sizes were generally smaller.

For EN, lesions were predominantly located in the tail (31.1%) and comprised of 9.7% IPMNs, 3% SCNs, 3.7% SPNs, and 92.8% PanNETs. The median tumor size was 1.7 cm. Conversion to oEN occurred in 2.5% patients and was due to technical difficulty or adhesions. One (0.6%) rEN patients underwent relaparotomy. Only one patient required an intraoperative transfusion.

Similarly, for CP, lesions were predominantly located in the neck/body (100%) and comprised of 19.7% IPMNs, 34.1% SCNs, 11.1% MCNs, 18.8% SPNs and 12.3% PanNETs. The median tumor size was 2.6cm. Two (0.6%) patients required conversion to oCP. Eleven (3.2%) patients needed relaparotomy, primarily to manage postoperative bleeding. No patients required intraoperative transfusions. For the rCP subgroup, the distal stump was reconstructed using a pancreato-gastrostomy anastomosis in 8.5% of cases, while a PJ was used in 72.9% of cases. In one series, an end-to-end pancreato-pancreatic anastomosis was used for the reconstruction.

Overall, the mean operative time for rPSP was 169.3 min. The mean operative time for rEN and rCP was 149.2 min and 177.7 min, respectively. Among studies that compared rPSP to open PSP, rPSP had a shorter operative time. Estimated blood loss (EBL) was 108.3 mL in the overall rPSP group and 1149.2 mL and 88.9 mL in the rEN and CP groups respectively. When compared to oEN and oCP, rEN and rCP had significantly lower EBL.

### 3.4. Early Postoperative Outcomes

The overall LOS was 16.9 days, with a LOS of 14.5 days for rEN and 17.6 days for rCP, respectively. Among studies that compared rPSP to oPSP, LOS was also shorter for the rPSP group. Major complications occurred in 5.4% of cases in the rPSP group. Specifically, clinically significant POPF occurred in 29.8% of cases. Among all included patients, zero events of postoperative mortality occurred.

### 3.5. Long-Term Postoperative Outcomes (Endo/Exo)

The median follow-up for rPSP patients was 24.1 months. Disease recurrence was inconsistently reported and was therefore not included in this review. During the follow-up period, exocrine deficiency occurred in 0.54% of cases in the rPSP group, 0% in the rEN subgroup, and 0.7% in the rCP subgroup. Similarly, endocrine deficiency occurred in 2.9%, 0%, and 5% in the rPSP, rEN, and rCP groups, respectively. The outcomes of case reports/series and comparative studies are summarized in Table 3 and Table 4, respectively. The outcomes stratified by procedure and approach are listed in Table 5.

## 4. Discussion

In this systematic review, we find that rPSP is commonly performed in the treatment of benign or indolent pancreatic lesions such as SPN and PanNET. This is the first review to focus on robotic parenchymal sparing procedures. We find that the robotic approach led to no deaths and was rarely linked to severe complications such as reoperation for bleeding. However, postoperative pancreatic fistula was common after rPSP and occurred in over 30% of all cases, but all were grade A or B and did not require reoperation. When compared to the open approach for similar indications, rPSP led to a longer average operative time, shorter length of stay, and higher estimated intraoperative blood loss. Postoperative endocrine and exocrine insufficiency were nearly nonexistent after rPSP. As more cystic and premalignant lesions of the pancreas are incidentally discovered with modern imaging, it is important to consider short-term disability and long-term function after pancreatic surgery. The robotic approach may allow patients to better realize the long-term benefits of parenchymal preservation with a nearly negligible risk of diabetes and minimal effect on the long-term quality of life. This is particularly important when considering rPSP for patients with SPN and PanNET, which are younger patients with a normal life expectancy [31,32]. However, short-term complications such as clinically significant POPF are common, and patients should be well-counseled about the risks and benefits of rPSP.

J.M.T. Finney first described a central pancreatectomy at Johns Hopkins as early as 1910, and since then, many other non-anatomic approaches have been developed [33]. These parenchymal-sparing operations—enucleation, uncinate resection, DPPPHR, and central pancreatectomy—all have a higher risk of early complications such as POPF when compared to standard anatomical resections (pancreatoduodenectomy or distal pancreatectomy). POPF occurs in up to 41% of patients undergoing enucleation, with half of those being significant enough to require intervention [34]. This is in comparison to an approximately 10–20% rate of clinically relevant POPF in patients undergoing pancreatoduodenectomy [35]. However, central pancreatectomy also leads to a lower rate of exocrine and endocrine insufficiency. The occurrence of postoperative endocrine insufficiency, intraoperative blood loss, and length of stay were all also lower with enucleation in a meta-analysis [36]. Similarly, in a randomized study by Büchler et al., patients undergoing DPPPHR had less pain, better glucose tolerance, and more rapid weight gain after surgery than those undergoing pancreatoduodenectomy [37]. Altogether, we find that POPF remains common after rPSP, although severe complications, endocrine/exocrine insufficiency, and reintervention are exceedingly rare. Parenchymal-sparing approaches prioritize long-term outcomes at the cost of temporary short-term morbidity, and patients should be acutely aware of these risks before committing to these operations.

When comparing rPSP to open PSP, we found no significant differences in overall morbidity using a robotic approach. The largest meta-analysis of 1004 patients undergoing minimally invasive central pancreatectomy showed no difference in the rate of severe complications, POPF, or new endocrine insufficiency with a minimally invasive approach [38]. Endocrine and exocrine functions are still preserved one year after a minimally invasive central pancreatectomy [39]. Similarly, minimally invasive enucleations were associated with a significantly decreased overall complication and POPF rate compared to similar open approaches [40]. Large studies of rPSP are still rare, but a meta-analysis of 13 studies of robotic central pancreatectomy similarly showed an overall complication rate of 58%, POPF in 41%, and a negligible rate of endocrine insufficiency [41]. A robotic approach may also lead to better splenic vessel preservation and fewer conversions to open surgery when compared to laparoscopic enucleations [40]. The benefits seen with rPSP over laparoscopic and open approaches may be due to enhanced visualization and dexterity afforded by the robotic platform, allowing for more careful dissection between small tumors and the pancreatic duct. These advantages seem common to all rPSP procedures, although the learning curve for robotic pancreatic resections remains steep.

Although the learning curve for rPSP approaches has yet to be defined, there is evidence that proficiency for robotic pancreaticoduodenectomy occurs after a minimum of 40 cases [42]. Among the cohort studies and randomized trials included in our review, we find that results are mixed with regard to improvement with rPSP over time. The most recent study on robotic enucleation with enough patients to make meaningful comparisons is from Caruso et al. in 2022, who found no severe complications or grade B/C pancreatic fistulas in their cohort of 40 patients, whereas prior studies had several patients with such complications [21]. On the other hand, Shi et al. published the largest and most recent series of central pancreatectomy in 2020, and they found a higher rate of grade B/C POPF (34.5% vs. 21.7%) in their patients undergoing robotic vs. open central pancreatectomy [26]. Although parenchymal-sparing approaches are technically challenging, authors have pointed to the improved visualization and dexterity with the robot and the use of pancreaticogastrostomy to decrease the need for a Roux-en-y reconstruction after central pancreatectomy [13] as reasons why rPSP approaches may actually be relatively easy to adopt.

This review has several limitations. Firstly, the procedures included together are technically very different, with disparate risk profiles. However, they do share POPF as a common and oft-feared complication, and this does not seem to be significantly elevated in patients undergoing rPSP. Of the studies included, there was only one randomized trial [22]; there were many case series and case reports among the remainder. As such, most studies were likely subject to a significant selection bias wherein we would expect healthier patients with more accessible tumors and favorable anatomy to undergo robotic surgery. In the studies included in this review, patients were mostly similar with regard to lesion size, BMI, age, and baseline ASA, but certain studies had significantly younger patients [24], smaller tumors [25], and healthier patients in their robotic cohort [13]. Furthermore, patients undergoing rPSP have been highly selected and include very few cases of malignancy, i.e., pancreatic ductal adenocarcinoma, and so these results may not be applicable to this subset of patients for which the majority of pancreatectomies are performed. However, these promising results for rPSP demonstrate that good results are achievable for selected patients in the hands of highly-trained individuals. These results may one day be more generalizable as these techniques are more widely adopted in practice and outcomes improve.

## 5. Conclusions

Ultimately, rPSP appears to be a safe approach for benign or premalignant pancreatic pathology. These findings show that rPSP may be effective for the management of non-malignant cystic pancreatic lesions, preserving pancreatic function and maximizing long-term benefits. Further research with larger sample sizes from high-volume centers of excellence in hepatopancreatobiliary surgery is needed before these techniques are widely adopted, as implementation has been thus far limited to a handful of centers with very carefully selected patients and demonstrating that these procedures can be performed safely at a larger scale is essential.

## Figures and Tables

**Table 1 cancers-15-04369-t001:** Case reports and case series of robotic parenchymal-preserving pancreatectomy.

Author	Country	Year	Study Type	Surgery Type	Total Patients
Boggi et al. [14]	Italy	2010	Case Series	CP	3
Guilianotti et al. [16]	Italy	2010	Case Series	CP	3
Addeo et al. [7]	France	2011	Case Report	CP	1
Peng et al. [17]	China	2012	Case Series	DPPPHR	4
Abood et al. [13]	USA	2013	Case Series	CP	9
Liang et al. [10]	China	2018	Case Report	Enucleation	1
Di Benedetto et al. [15]	Italy	2019	Case Series	Enucleation	12
Wang et al. [18]	China	2019	Case Series	CP	11
Chong et al. [8]	Republic of Korea	2019	Case Report	Enucleation	1
Machado et al. [11]	Brazil	2019	Case Report	Uncinectomy	1
Ku et al. [9]	Republic of Korea	2020	Case Report	CP	1
Van Ramshorst et al. [12]	The Netherlands	2021	Case Report	CP	1

CP: central pancreatectomy; DPPPHR: duodenum-preserving pancreatic head resection.

**Table 2 cancers-15-04369-t002:** Retrospective cohort studies and randomized trials of robotic parenchymal-preserving pancreatectomy.

Author	Country	Year	Study Type	Comparison	Patients		MINORS Score
					Group 1	Group 2	
Kang et al. [24]	Korea	2011	Retrospective	Robotic vs. Open CP	5	10	16
Cheng et al. [19]	China	2012	Retrospective	Robotic vs. Open CP	7	36	16
Zhang et al. [29]	China	2015	Retrospective	RCP in Elderly vs. Young Patients	10	55	15
Jin et al. [23]	China	2016	Retrospective	Robotic vs. Open EN	31	25	16
Shi et al. [25]	China	2016	Retrospective	Robotic vs. Open EN	26	17	16
Tian et al. [27]	China	2016	Retrospective	Robotic vs. Open EN	60	60	14
Chen et al. [22]	China	2017	RCT	Robotic vs. Open CP	50	50	20
Bartolini et al. [20]	Italy	2019	Retrospective	Robotic ENs vs. Robotic Whipple/DP	9	16	16
Shi et al. [26]	China	2020	Retrospective	Robotic vs. Open CP	110	60	16
Wang et al. [28]	China	2021	Retrospective	Robotic End-to-End Pancreatic Reconstruction vs. PJ	52	22	16
Caruso et al. [21]	Spain	2022	Retrospective	Robotic vs. Open EN	20	20	15

CP: central pancreatectomy; DP: distal pancreatectomy; EN: enucleation; MINORS: methodological Index for Non-randomized Studies; PJ: pancreaticojejunostomy.

**Table 3 cancers-15-04369-t003:** Outcomes of case reports and case series of robotic parenchymal-preserving pancreatectomy.

Author	Case Length (min)	EBL (mL)	Convert Open (%)	Morbidity		POPF		Length of Stay (Days)	Follow Up (Mos)	Exocrine Insufficiency	Endocrine Insufficiency
				Total (%)	CD ≥ 3 (%)	A (%)	B/C (%)				
Abood [13]	415.3 ± 67	195 ± 100	1 (11.1)	7 (77.8)	1 (11.1)	5 (55.6)	2 (22.2)	11.5 ± 4	1	0	0
Boggi [14]	426.7 *	Φ	0	2 (66.7)	1 (33.3)	1 (33.3)	1 (33.3)	14.3 ± 10	27	0	0
Di Benedetto [15]	203.17 *	38.3 *	0	4 (33.3)	1 (8.3)	2 (16.7)	1 (8.3)	3.9 *	17	0	0
Guilianotti [16]	320 *	233 *	0	1 (33.3)	0	0	1 (33.3)	15 ± 10	47	0	0
Peng [17]	298.8 ± 34	425 ± 236	0	3 (75)	0	0	3 (75)	26.8 ± 5	Ω	0	0
Wang [18]	121 *	55 ± 25	0	7 (63.6)	1 (9.1)	1 (9.1)	1 (9.1)	6.5 ± 1	11.7	0	0
Addeo [17]	450	300	0	1 (100)	0	1 (100)	0	15	Ω	Φ	Φ
Chong [8]	124	50	0	0	0	0	0	4	Ω	Φ	Φ
Ku [9]	295	50	0	1 (100)	0	1 (100)	0	9	Ω	Φ	Φ
Liang [10]	65	5	0	0	0	0	0	6	18	0	0
Machado [11]	215	50	0	1 (100)	0	0	1 (100)	3	0.5	Φ	Φ
Van Ramshorst [12]	248	20	0	1 (100)	0	0	1 (100)	8	Ω	0	0

* Standard deviation was not provided and could not be extrapolated. Φ: not reported. Ω: not followed. EBL: estimated blood loss; CD: Clavien–Dindo grade; POPF: postoperative pancreatic fistula.

**Table 4 cancers-15-04369-t004:** Outcomes for retrospective cohort studies and randomized trials of robotic parenchymal-preserving pancreatectomy.

Author	Case Length (min)	EBL (mL)	Transf.	Convert Open	Reop.	Morbidity		POPF		Length of Stay (Days)	Follow Up (Mos)	Exocrine Insuff.	Endocrine Insuff.
						None	CD ≥ 3	A	B/C				
Enucleation (Robotic v. Open)													
Caruso (2022) [21]	210 ± 78 vs. 180 ± 76	Φ	0 vs. 0	1 vs. 0	0 vs. 1	17 vs. 12	Φ	2 vs. 3	0 vs. 0	8.3 ± 1 vs. 13.8 ± 2	12 vs. 12	Φ	Φ
Jin (2016) [23]	103.3 ± 23.3 vs. 148.7 ± 62.9	30 ± 31.1 vs. 127.7 ± 143.9	0 vs. 0	0 vs. 0	0 vs. 1	25 vs. 16	2 vs. 4	11 vs. 8	12 vs. 13	15.7 ± 11 vs. 20.7 ± 13	19.1 vs. 14.8	0 vs. 0	0 vs. 0
Shi (2016) [25]	124.6 ± 50.9 vs. 198.5 ± 70.7	76 ± 85.4 vs. 157.1 ± 114.2	0 vs. 0	0 vs. 0	0 vs. 0	14 vs. 9	0 vs. 0	5 vs. 4	7 vs. 3	22.6 ± 16 vs. 23.9 ± 17	25 vs. 25	0 vs. 0	0 vs. 0
Tian (2016) [27]	148 ± 57.5 vs. 170 ± 43.2	268.8 ± 214 vs. 345 ± 255.1	1 vs. 1	3 vs. 0	0 vs. 0	52 vs. 44	2 vs. 6	Φ	6 vs. 10	16.5 ± 7 vs. 30.5 ± 18	3 vs. 3	Φ	Φ
Central Pancreatectomy (Robotic vs. Open)													
Cheng (2013) [19]	181.8 ± 107.4 vs. 221 ± 54.9	212.5 ± 128.3 vs. 487.5 ± 342.9	0 vs. 5	0 vs. 0	0 vs. 0	1 vs. 18	Φ	0 vs. 9	5 vs. 6	22 ± 7 vs. 38.5 ± 23	23 vs. 62	0 vs. 0	0 vs. 3
Chen (2017) [22]	162.5 ± 20.1 vs. 194 ± 15.6	62.5 ± 11.1 vs. 198.8 ± 45.7	0 vs. 5	0 vs. 0	2 vs. 2	27 vs. 30	7 vs. 9	13 vs. 9	9 vs. 18	15.6 ± 5 vs. 21.7 ± 13	Φ	Φ	Φ
Kang (2011) [24]	432 ± 65.7 vs. 286.5 ± 90.2	275 ± 221.7 vs. 858.3 ± 490	0 vs. 3	0 vs. 0	0 vs. 2	4 vs. 5	1 vs. 0	4 vs. 0	0 vs. 1	14.6 ± 8 vs. 22.1 ± 13	19 vs. 19	Φ	0 vs. 0
Shi (2020) [26]	162 ± 63 vs. 208 ± 52	88 ± 93 vs. 195 ± 165	Φ	0 vs. 0	5 vs. 2	53 vs. 36	0 vs. 0	Φ	38 vs. 13	24.5 ± 13 vs. 23.3 ± 18	54 vs. 54	0 vs. 0	3 vs. 3

Φ: not reported. CD: Clavien–Dindo grade; EBL: estimated blood loss; Insuff: insufficiency; POPF: postoperative pancreatic fistula; Transf: transfusion.

**Table 5 cancers-15-04369-t005:** Postoperative stratified by procedure type and approach.

	rPSP	rEN	rCP	oPSP	oEN	oCP
Mean age (years)	48.9	50.1	48.6	50.6	50.3	50.8
Mean BMI (kg/m^2^)	23.7	25.2	23	23.7	25.1	22.2
Mean lesion size (cm)	2.3	1.7	2.6	2.6	2	3
Mean OR time (min)	169.3	146.7	177.7	193.9	171.3	211.5
Mean EBL (mL)	108.3	149.2	88.9	288.1	260.4	306.2
Mean LOS (days)	16.9	14.5	17.6	25.6	14.1	26.2
Male (%)	188 (37.2%)	66 (41.3%)	120 (35.3%)	104 (37.4%)	50 (41%)	54 (34.6%)
ASA > 3 (%)	19 (8.8%)	8 (11.8%)	11 (7.5%)	6 (6.9%)	4 (10.8%)	2 (4%)
Location						
Head	54 (10.9%)	40 (29.8%)	0	36 (13.8%)	36 (34.3%)	0
Uncinate	23 (4.6%)	11 (8.2%)	0	13 (5%)	13 (12.4%)	0
Neck/body	374 (75.4%)	27 (36.5%)	331 (100%)	164 (81.6%)	8 (17.8%)	
Tail	39 (7.9%)	23 (31.1%)	0	18 (8.9%)	18 (40%)	156 (100%)
Pathology						
IPMN	74 (15.4%)	13 (9.7%)	61 (17.9%)	44 (16.8%)	11 (10.5%)	0
SCN	121 (25.3%)	4 (3%)	116 (34.1%)	36 (13.8%)	2 (1.9%)	36 (23.1%)
MCN	47 (9.8%)	0 (0%)	46 (13.5%)	29 (11.1%)	4 (3.8%)	25 (16%)
SPN	69 (14.4%)	5 (3.7%)	64 (18.8%)	19 (7.3%)	2 (1.9%)	17 (10.9%)
PanNET	155 (32.36%)	111 (82.8%)	42 (12.3%)	111 (42.5%)	86 (81.9%)	25 (16%)
Complications						
Conversion to open	6 (1.2%)	4 (2.5%)	2 (0.6%)	0	0	0
Reoperation	12 (2.4%)	1 (0.6%)	11 (3.2%)	8 (2.9%)	2 (1.6%)	6 (3.8%)
CD > 3	26 (5.4%)	6 (4.3%)	20 (6%)	119 (8.5%)	10 (9.8%)	9 (7.5%)
CS POPF	149 (29.8%)	26 (16.3%)	119 (35.5%)	64 (23%)	26 (21.3%)	38 (24.3%)
Endocrine insufficiency	9 (2.9%)	0	9 (5%)	3 (3.4%)	0	3 (6.5%)
Exocrine insufficiency	2 (0.54%)	0	2 (0.7%)	0	0	0
Anastomosis						
Pancreatogastrostomy	29 (6.8%)	0	29 (8.5%)	91 (41.7%)	0	91 (58.3%)
Pancreatojejunostomy	251 (59.6%)	0	248 (72.9%)	65 (29.8%)	0	65 (41.6%)
Pancreatopancreatic	63 (17.2%)	0	63 (18.5%)	0	0	0

## Data Availability

Publicly available studies were used in this systematic review. These can be found in PubMed (Medline) and Embase.

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
