# Peer review of "Robotic Parenchymal-Sparing Pancreatectomy: A Systematic Review"

_cancers, 2023, doi:10.3390/cancers15174369_

Round 1

Reviewer 1 Report

Cancers -2535588

Comments and Suggestions for Authors:

This review study was aimed to investigate early postoperative outcome after robotic pancreatic sparing pancreatectomy for various borderline and benign tumors. There have been still very few reports about surgical results of robotic pancreatic sparing pancreatectomy at the present time. Therefore, it may be still too early to perform the systematic review nowadays. Most reported literatures were case reports and case series from one and 12 cases, and comparative studies of robotic parenchyma-sparing pancreatectomy were investigated to compare with open procedures rather than laparoscopic approach.

1, The parenchymal sparing pancreatectomy has been reported until the present time that these approaches could bring about clinically relevant POPF so frequently. Therefore, the parenchyma sparing pancreatectomy has not been always recognized as the standard procedure for various localized lesions by most pancreatic surgeons over the world, even by open approach. Laparoscopic and also robotic approach could not overcome the frequent POPF complication after parenchyma preserving pancreatectomy.  Why could authors give the conclusion that robotic parenchyma sparing pancreatectomy appears to be safe according these results in conclusion? I think that it does not make sense scientifically.

2, This review manuscript mayt give unjustified risky message that the parenchyma preserving pancreatectomy for various lesions could be an useful choice of surgical strategy. Authors from high-volume center of HPB surgery should evaluate the usefulness of these approach more carefully at the present time when there were insufficient volumes of reported literatures.  

3, In Table 1, “Central” of surgical type by Guilianotti was unclear. Did it mean central pancreatectomy?

Author Response

Comment: This review study was aimed to investigate early postoperative outcome after robotic pancreatic sparing pancreatectomy for various borderline and benign tumors. There have been still very few reports about surgical results of robotic pancreatic sparing pancreatectomy at the present time. Therefore, it may be still too early to perform the systematic review nowadays. Most reported literatures were case reports and case series from one and 12 cases, and comparative studies of robotic parenchyma-sparing pancreatectomy were investigated to compare with open procedures rather than laparoscopic approach.

1, The parenchymal sparing pancreatectomy has been reported until the present time that these approaches could bring about clinically relevant POPF so frequently. Therefore, the parenchyma sparing pancreatectomy has not been always recognized as the standard procedure for various localized lesions by most pancreatic surgeons over the world, even by open approach. Laparoscopic and also robotic approach could not overcome the frequent POPF complication after parenchyma preserving pancreatectomy.  Why could authors give the conclusion that robotic parenchyma sparing pancreatectomy appears to be safe according these results in conclusion? I think that it does not make sense scientifically.

Response: Thank you for the commentary. In the discussion, we mention that although grade B POPF is common after parenchymal sparing pancreatecomy, severe morbidity (i.e. reoperation, mortality) is rare, and that although short-term complications are exceedingly common, long-term complications are not. However, we agree that there is still no high-level evidence to support the adoption of rPSP globally. We have toned down the language of the abstract, discussion, and conclusion to be more specific and careful about the adoption of parenchymal-sparing approaches.  

Comment: 2, This review manuscript may give unjustified risky message that the parenchyma preserving pancreatectomy for various lesions could be an useful choice of surgical strategy. Authors from high-volume center of HPB surgery should evaluate the usefulness of these approach more carefully at the present time when there were insufficient volumes of reported literatures. 

Response: We have changed the language of the discussion and conclusion to be more nuanced in our recommendations regarding adopting parenchymal-preserving pancreatectomy (see the previous response above). We have also specifically added this recommendation that research from high-volume HPB centers is necessary to the conclusion. 

Comment: 3, In Table 1, “Central” of surgical type by Guilianotti was unclear. Did it mean central pancreatectomy?

Response: Yes, this has been changed to say “CP” which is consistent with the other central pancreatectomies in the same table.

Reviewer 2 Report

I appreciate the opportunity to review this manuscript. I want to commend the authors. I found the methodology sound and the overall question important and useful.

I have a couple of comments for the authors to consider. These may be more for editorial comment but perhaps some details are available in the literature that they reviewed.

First, I suspect that there is some operator dependence to consider. Both robotic surgery and pancreatic surgery are, as individual entities, highly skilled techniques and the marriage of the two seems even more so. Am I correct in this assumption and how does this impact outcomes, which seem otherwise to be terrific with this approach?

Second, it is good to see that the articles considered in this review spanned a stretch of time. Has technique improved during this 12 year span; are outcomes better in more recent years as a result?

Finally, given the manuscripts considered, I would agree with the conclusion focusing on the types of tumors included for application. It would seem a much larger scope but also an ideal one would be to extend to PDAC. That has other issues of course, including peri-operative therapy need, but might be worthwhile to acknowledge that consideration in a sentence or so.

Author Response

Comment: I appreciate the opportunity to review this manuscript. I want to commend the authors. I found the methodology sound and the overall question important and useful.

I have a couple of comments for the authors to consider. These may be more for editorial comment but perhaps some details are available in the literature that they reviewed.

First, I suspect that there is some operator dependence to consider. Both robotic surgery and pancreatic surgery are, as individual entities, highly skilled techniques and the marriage of the two seems even more so. Am I correct in this assumption and how does this impact outcomes, which seem otherwise to be terrific with this approach?

Response: In the 3rd paragraph of the discussion, we discuss the added morbidity of performing a robotic parenchymal-sparing approach compared to open techniques. To summarize, we found that severe complications after parenchymal-sparing approaches were rare, and that there was no significant increase in morbidity with the addition of the robotic approach, specifically with regards to pancreatic fistula, insufficiency, and severe complications. We have made some changes to the phrasing of this paragraph in order to make it clear that we are looking at the added morbidity from robotic approaches.

Comment: Second, it is good to see that the articles considered in this review spanned a stretch of time. Has technique improved during this 12 year span; are outcomes better in more recent years as a result?

Response: The learning curve for robotic parenchymal-sparing approaches has never before been characterized, but there is data on the learning curve for robotic pancreatectomy (i.e. Whipple, distal) which we have now included in the end of the 3rd paragraph of the discussion. With regards to improvement over time within individual studies, the majority of these studies are still very small and the authors do not comment on whether or not their outcomes improved towards the end of their study period. When looking at improvement between different studies over time, results are mixed. The most recent study on robotic enucleation with enough patients to make meaningful comparisons is from Caruso et al. in 2022, and they found no severe complications or grade B/C pancreatic fistulas in their cohort of 40 patients whereas prior studies had several patients with such complications. On the other hand, Shi et al. published the largest and most recent series of central pancreatectomy in 2020, and they found a higher rate of grade B/C POPF (34.5% vs 21.7%) in their patients undergoing robotic vs. open central pancreatectomy. This has been added to the discussion as the 4th paragraph.

Comment: Finally, given the manuscripts considered, I would agree with the conclusion focusing on the types of tumors included for application. It would seem a much larger scope but also an ideal one would be to extend to PDAC. That has other issues of course, including peri-operative therapy need, but might be worthwhile to acknowledge that consideration in a sentence or so.

Response: We acknowledge the fact that most of these patients undergoing rPSP have been highly selected and do not include many patients with PDAC. This has been added to the last paragraph of the discussion.

Reviewer 3 Report

Good review. Plain laparoscopic approach should not be excluded.

Author Response

Comment: Good review. Plain laparoscopic approach should not be excluded.

Response: Thank you for the feedback. We did not find many studies discussing laparoscopic parenchymal-preserving approaches; likely, this is because these techniques are technically challenging without the technical advantages of the robotic platform. For the sake of making a straightforward comparison between groups, we did not want to focus on this small subset of laparoscopic patients.

Round 2

Reviewer 1 Report

Cancers-2535588

Comments and Suggestions for Authors;

The revised manuscript was almost appropriately changed according to all reviewer’s comments.

Especially the clinical role of parenchymal-sparing pancreatectomy for benign and borderline pancreatic lesions as one of choice among required therapeutic surgical procedures was rewritten to be nuanced in their recommendation regarding adopting parenchymal-preserving pancreatectomy.

The revised manuscript could be acceptable for the publication.